:ᗺ∕: PLOS | ONE

# Evaluation of a global spring wheat panel for stripe rust: Resistance loci validation and novel resources identification

Ibrahim S. Elbasyoni [1,2]*, Walid M. El-Orabey[3], Sabah Morsy[1], P. S. Baenziger[2], Zakaria Al Ajlouni[4], Ismail Dowikat[2]

**1** Crop Science Department, Damanhur University, Damanhur, Egypt, **2** Department of Agronomy and Horticulture, University of Nebraska, Lincoln, NE, United States of America, **3** Wheat Diseases Res. Department, Plant Pathology Res. Institute, ARC, Giza, Egypt, **4** Jordan University of Science and Technology, Department of Plant Pathology, Irbid, Jordan

* ielbasyoni2@unl.edu

**Data Availability Statement:** Data has been uploaded as Supporting Information and to https://triticeaetoolbox.org/wheat/.

## Abstract

Stripe rust (incited by *Puccinia striiformis* f. sp. *tritici*) is airborne wheat (*Triticum aestivum* L.) disease with dynamic virulence evolution. Thus, anticipatory and continued screening in hotspot regions is crucial to identify new pathotypes and integrate new resistance resources to prevent potential disease epidemics. A global wheat panel consisting of 882 landraces and 912 improved accessions was evaluated in two locations in Egypt during 2016 and 2017. Five prevalent and aggressive pathotypes of stripe rust were used to inoculate the accessions during the two growing seasons and two locations under field conditions. The objectives were to evaluate the panel for stripe rust resistance at the adult plant stage, identify potentially novel QTLs associated with stripe rust resistance, and validate previously reported stripe rust QTLs under the Egyptian conditions. The results indicated that 42 landraces and 140 improved accessions were resistant to stripe rust. Moreover, 24 SNPs were associated with stripe rust resistance and were within 18 wheat functional genes. Four of these genes were involved in several plant defense mechanisms. The number of favorable alleles, based upon the associated SNPs, was significant and negatively correlated with stripe rust resistance score, i.e., as the number of resistances alleles increased the observed resistance increased. In conclusion, generating new stripe rust phenotypic information on this panel while using the publicly available molecular marker data, contributed to identifying potentially novel QTLs associated with stripe rust and validated 17 of the previously reported QTLs in one of the global hotspots for stripe rust.

## Introduction

Wheat (*Triticum aestivum* L.) stripe (syn. yellow) rust caused by *Puccinia striiformis* f. sp. *tritici* (Pst), is one of the most devastating wheat diseases in the world. Stripe rust can cause yield losses from 10 to 100% [1]. Utilization of rust-resistant genotypes is the most economical and environmentally sound approach to reduce stripe rust damage, as it protects grain yield and

**Funding:** This study was supported by STDF grant#14935 to ISE.

**Competing interests:** The authors have declared that no competing interests exist.

reduces the need for fungicides [2]. Global efforts dedicated to identifying stripe rust-resistance genes, resulted in 74 officially designated and more than 40 temporarily named stripe rust-resistant genes [2–4]. Most of the identified stripe rust resistance genes (*Yr*)are race-specific resistance genes [5] and thus vulnerable to the evolving *Yr* pathotypes. Stripe rust evolves new pathotypes quickly through mutation and somatic hybridization [6], and because it is air-borne, races can migrate to other regions and become regionally or globally predominant [7]. For this reason, pathotype non-specific resistance is considered more durable and effective against many stripe rust pathotypes. For example, *Yr18* has remained durable and effective against stripe rust for 50 years [8]. Therefore, breeding durable stripe rust-resistant genotypes remains one of the key objectives for wheat pathologists and breeders.

Identifying adult plant stripe resistant genotypes is often done in the field due to the limitations of growing adult plants in controlled environments. However, screening under the field conditions requires the presence of the pathogen and conducive conditions for infection [9]. Also, field disease screening is affected by the seasonal variability of temperature and precipitation, which in return affect the susceptibility of the host [10]. Therefore, it is often desirable to artificially inoculate the host plant under field conditions to assure the presence of common races in the region [11]. To evaluate plant materials for new stripe rust pathotype(s) that is not naturally present in a breeder's environment; artificial inoculation is not acceptable as it will release new pathotypes into the environment. In this case, evaluation should be undertaken at the rust hotspots, e.g., where the new pathotypes naturally occur [1]. For example, the world-wide wheat collections were screened for stripe rust resistance in Pakistan [12] and India [13], both of which are considered hotspot regions for this disease.

Field evaluation is expensive, time-consuming and, as mentioned earlier, highly affected by environmental conditions. The advent of relatively inexpensive, high throughput molecular marker platforms makes the marker-assisted selection (MAS) a viable approach to tracking resistance genes. In MAS, DNA molecular markers are used to select for desirable traits based on previous knowledge of the association between a specific marker and that trait. Therefore, establishing a marker-trait relationship is the first step in developing MAS protocols for any given trait. The two requirements to build the marker-trait association are to have accurate phenotypic information and reliable marker data. Furthermore, establishing accurate marker-trait associations require large populations to obtain a higher power by increasing the recombination frequency and the frequency of rare alleles [14]. Maccaferri et al. (2015) [15] evaluated 1000 spring wheat accessions using four stripe rust pathotypes. They were able to identify 97 SNP linked with stripe rust resistance. Muleta et al. (2017) [16] evaluated 1,163 spring wheat accessions for stripe rust resistance and were able to identify 11 and 7 genomic regions in significant associations with stripe rust resistance at the adult and seedling stages, respectively. Kertho et al. (2015) [17] evaluated 567 winter wheat accessions for the stripe and leaf rust (*Puccinia triticina* Eriks) resistance and identified 65 and eight significant markers associated with leaf rust and stripe rust, respectively. These studies identified markers or QTLs associated with stripe rust resistance. Such markers, after validation, will be useful in breeding resistant genotypes, which will save time and resources [18].

Currently, accurate phenotyping has become the major bottleneck and funding constraint of MAS applications [14]. Thus, we focused our efforts and resources on conducting extensive phenotyping of global wheat collection under the field conditions in Egypt. The five prevalent and aggressive pathotypes in Egypt during the last five years of stripe rust are 0E0, 6E4, 70E20, 128E28, and 134E244 [19]. Several of the most important wheat cultivars such as "Misr2", "Giza168" and "Sakha 61" known to be resistant to the previous five most important pathotypes, recently have become susceptible under the field conditions in Egypt. During the last 10 years, stripe rust races have evolved in Egypt and became more prevalent and aggressive. For

example, infections of stripe rust were observed on several lines with stripe rust resistance genes, i.e., *Yr1*, *Yr17* and *Yr32* in northern Egypt during 2012 [20]. Aggressive virulent pathotypes to Yr27 were detected on Yr27 resistant lines such as Yr27/6*Avocet S, and Ciano 97. Additionally, during 2015, the warrior pathotype [21,22] (virulent on Yr1, Yr2, Yr3, Yr4, Yr6, Yr7, Yr9, Yr17, Yr25, Yr32, and YrSp) was detected.

Wheat accessions used in this study were obtained from several geographic regions and included landraces, experimental lines, and cultivars. Most of these lines have not been tested for stripe rust resistance in Egypt, one of the world hotspots for stripe rust [23]. Thus, testing this large number of accessions in Egypt should add an important regional perspective to the previous studies that were conducted using the same plant materials in other regions. Furthermore, in the current study, we used the same SNP markers platform (9K SNP) that was used in recent stripe rust studies [15,16,17]. Thus, phenotypic information generated from the current study coupled with the genotypic data can be used to identify new or environmental specific (local) QTLs and to validate recently reported stripe rust QTLs under the Egyptian environmental conditions. The objectives of this study were to 1- Evaluate a comprehensive spring wheat collection for stripe rust resistance during the adult plant growth stage to identify new resistant genotypes, 2- Identify potential QTLs associated with stripe resistance, and 3- Validate previously identified stripe rust associated QTLs or identify new QTLs under the Egyptian field conditions.

## Materials and methods

### Ethics statement

The fields used in the current study located within commercial wheat production regions in Egypt. Additionally, yellow rust races used in the study were among the most common races in these regions. Thus, we did not introduce any race that is not naturally present in the growing environment. Furthermore, any field activities were conducted properly within the Egyptian laws and regulations by an Agriculture research center (ARC) specialist (Second author on this paper). Therefore, no specific permissions were required for locations or field activities. Furthermore, we confirm that the field studies conducted in the current study did not involve endangering indigenous or protected species.

### Plant materials and experimental conditions

The present study was conducted in two consecutive growing seasons (2015/2016 and 2016/2017; hereafter referred to by their harvest season, 2016, and 2017) and two locations in Behira governorate, Egypt, i.e., Elbostan, and Elkhazan. Hence the study was done in four environments (2 locations × 2 growing seasons). Elbostan location is an experimental farm for Damanhour University (30˚46′46″ N, 30˚82′32″ E), representing the newly reclaimed land, while Elkhazan location is a grower farm (31˚05'35.2"N, 30˚30'10.4"E) located in the Nile valley representing long-term farmed soil.

The seeds of all accessions were provided by the USDA-ARS, National Small Grains Collection (NSGC) located in Aberdeen, ID, USA. The accessions originated from 107 countries, including 35 accessions from Egypt and represented old and new accessions for the period from 1920 to 2012. The accessions included 882 landraces; 912 improved accessions (493 experimental lines and 419 cultivars) and 317 with unknown improvement category from a global spring wheat collection. Accessions details, i.e., pedigree, selection history, and origin can be found on the T3/wheat website (https://triticeaetoolbox.org/wheat/). Each accession was planted in two replicates using a randomized incomplete block design [24] in plots of four rows wide with 25 cm between rows and two meters long. The incomplete blocks consisted of

50 accessions in addition to three check cultivars, i.e., "Sids13", Gimmiza9, and Giza168. A spreader cultivar i.e., "Morocco" was planted around each replicate as a border of one meter wide. For field inoculation with stripe rusts, within each environment Morocco was sprayed with a mist of water and dusted with mixture of urediniospores at sunset, before dew formation, using 200 mg of five prevalent and aggressive pathotypes of stripe rust, i.e., 0E0, 6E4, 70E20, 128E28, and 134E244 [19] mixed with a talcum powder at a ratio of 1 : 20 (v/v) (spores: talcum powder). The inoculation of the spreader plants was conducted at the booting stage [25]. The urediniospores of the stripe rust were obtained from the Wheat Diseases Research Department, Plant Pathology Research Institute, Agricultural Research Center, Giza, Egypt. Standard agronomic practices including recommended fertilizer application and irrigation schedule were followed at each location.

## Disease assessment

Scoring stripe rust was conducted based on the Field Response (FR) fand the percentage of infected tissue (severity). The field responses used were Immune = 0, no uredinia or other macroscopic sign of infection, R = resistant, small uredinia surrounded by necrosis; MR = Moderately resistant, medium to large uredinia surrounded by necrosis; MS = moderately susceptible, medium to large uredinia surrounded by chlorosis; S = susceptible, large uredinia without necrosis or chlorosis [26]. To facilitate the statistical analysis, the field responses were converted into 0, 2, 4, 6 and 8 for immune, resistant, moderately resistant, moderately susceptible and susceptible, respectively for each replicate. Additionally, the genotypes that had across environments'average field response in the range from 1 to 3 were considered resistant; while those in the range from 3 to 5 were considered moderately resistant. Furthermore, the genotypes that had across environments'average field response in the range from 5 to 6 were considered moderately susceptible, and those fall in the range from 6 to 8 were considered susceptible.

## SNP genotyping

Wheat accessions included in this study were genotyped through the Triticeae Coordinated Agriculture Project (TCAP) using Illumina GoldenGate platform (Illumina Inc., San Diego, CA) at the USDA-ARS genotyping laboratory in Fargo, ND, USA [27]. Marker data were coded as x = {-1, 0, 1}, where -1 represents homozygous for the minor allele, 0 represents heterogeneous, and 1 represents homozygous for the major allele. The single nucleotide polymorphism (SNP) markers were filtered by removing SNPs with missing values > 10% and minor allele frequency [MAF] < 5%. The filtration step resulted in 3216 high-quality SNPs, missing values were imputed using random forest regression [28], which was applied using the MissForest R/package [29]. Filtered SNP markers were plotted in Manhattan plots using "wnsp 2013 consensus map"; available on: https://triticeaetoolbox.org/wheat/ [30].

## Statistical analysis

Analysis of variance was carried out by fitting the following model [31]:

$$Y_{ijlm} = \mu + E_i + EB_{(il)j} + G_m + EG_{im} + \varepsilon_{ijlm}$$

where $Y_{ijlm}$ is a vector of FR scores or ACI values measured on the $_{ijlm}$ plot, $\mu$ is the overall mean, $E_i$ is the effect of $i^{th}$ environment, $EB_{(i)j}$ is $j^{th}$ incomplete block nested within $_l^{th}$ complete block and $i^{th}$ environment (random), $G_m$ is the effect of $m^{th}$ genotype, $EG_{im}$ is the interaction effect between $i^{th}$ environment and $m^{th}$ genotype, and $\varepsilon_{ijlm}$ is the experimental error.

Homogeneity of variance across locations and growing seasons was tested using Bartlett's Test [32]. Means within and across locations were compared using Tukey's honest significant difference (HSD) [33].

The broad sense heritability (H$^2$) was estimated as follows:

$$H^2 = \frac{\sigma^2_G}{\sigma^2_G + \sigma^2_{G \times E} + \sigma^2_{error}}$$

Where $\sigma^2_G$ is the genetic variance; $\sigma^2_{GxE}$ is the genotype by environment variance and $\sigma^2_{error}$ is the residual variance.

Ensembling machine learning approach applied in the random forest algorithm was used to build a classification model using the accessions with known improvement class, i.e., improved or landraces, across the 3215 SNP markers (variables). SNP marker data on the accessions within the known improvement class was used to estimate the model parameters, which were then used to obtain a classification rule to group the accessions of the unknown improvement class into either improved or landraces [34]. Classification accuracy was estimated by randomly masking the improvement class for a set of accessions, and then using the random forest classification model to estimate the masked improvement class. The observed (previously known) and estimated improvement class was used to calculate the percentage of the misclassified accessions, as an average after replicating the previous process 100 times [35].

After classifying the unknown accessions, genetic variability among accessions within landraces and improved accessions was investigated using SNP markers by estimating pairwise allele sharing matrix among all accessions [36]. Eigenvector decomposition of the standardized allele sharing matrix was used to investigate the relationships among accessions within the improved lines and the landraces in which the first two principal components were plotted against each other while color coding accessions according to their improvement class using prcomp function in R software [37]. Then, a heatmap with a dendrogram was generated using heatmap.2 and hclust functions in R software [37] to visually identify the overall patterns in the studied materials. Polymorphic information content (PIC) was calculated according to Smith et al. (1997) [38].

The best linear unbiased estimates (BLUE) for stripe rust scores and SNP markers were subjected to association analysis using a mixed linear model (MLM) in R package GAPIT [39]. The association analysis was carried out by performing a linear mixed model with restricted maximum likelihood estimates as follows:

$$y = \mu + zu + Wm + e$$

Where Y is a vector of the stripe rust scores from the field responses, μ is a vector of intercepts, u is a vector of n×1 of random polygene background effects, e is a vector of random experimental errors with mean 0 and covariance matrix Var (e), Z is a matrix relating Y to u. Var(u) = 2KVg, where K is a known n×n matrix of realized relationship matrix, Vg is a scalar of the unknown genetic variance. m is a vector of fixed effect due to SNP markers, W is a matrix that relates Y to m. Var (e) = RVR, where R is an n×n matrix, and VR is scalar with unknown residual variance. P-values estimated from the association model were subjected to false discovery rate (FDR) corrections using Q-value estimates applied in the R package q-value [40]. The sequence of each significant SNP markers were used in the Triticeae Toolbox database to identify genes associated with YR and their functional annotation using IWGSC RefSeqv1.0 wheat reference genome [41].

## Results

### Genetic characterization of the studied accessions

Plotting the accessions using the first two principal components (Fig 1A) indicated that the accessions were grouped into two major groups. The first group contained the improved accessions while the second contained landraces. The unknown accessions were scattered across both groups implying that the unknown accessions included both improved accessions and landraces. The random forest model classified the unknown accessions (317) into improved (147) and landraces (170), with an estimated 91% classification accuracy (Fig 1B). The distribution of minor allele frequency (MAF), and the polymorphism information content (PIC) for the landraces and the improved accessions used in the current study are presented in Fig 2A and 2B, respectively. For landraces and improved accessions, PIC-values ranged from 0.09 to 0.37. The overall mean and median for PIC-values estimated from landraces were 0.28 and 0.30, respectively (Fig 2A). The PIC-value mean and median for the improved accessions were 0.30 and 0.33, respectively. The overall mean and median of MAF for the landraces were 0.26 and 0.29, respectively. The MAF mean and median for the improved accessions was 0.30 for both parameters (Fig 2B).

The distribution of the pair-wise shared alleles for the improved accessions and the landraces (Fig 3) indicated that the landraces tend to have less shared alleles compared to the improved accessions. The mean and median of the shared alleles among the landraces were 0.60 and 0.61, respectively. While the mean and median for the improved accessions were 0.69 and 0.70, respectively. Furthermore, the overall heatmap for the pairwise shared alleles in the landraces compared with the improved accessions indicated that the landraces tend to have fewer shared alleles (S1 Fig)

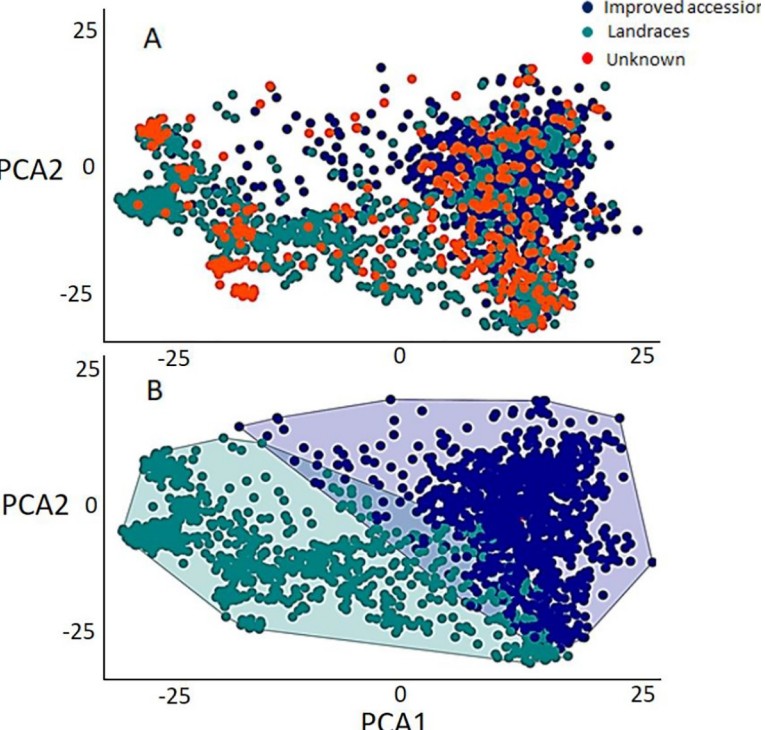

**Fig 1.** Accessions distribution based on the improvement degree, i.e, improved, landraces and unknown, using the first two principal components, before [A] and after [B] classifying the unknown accessions.

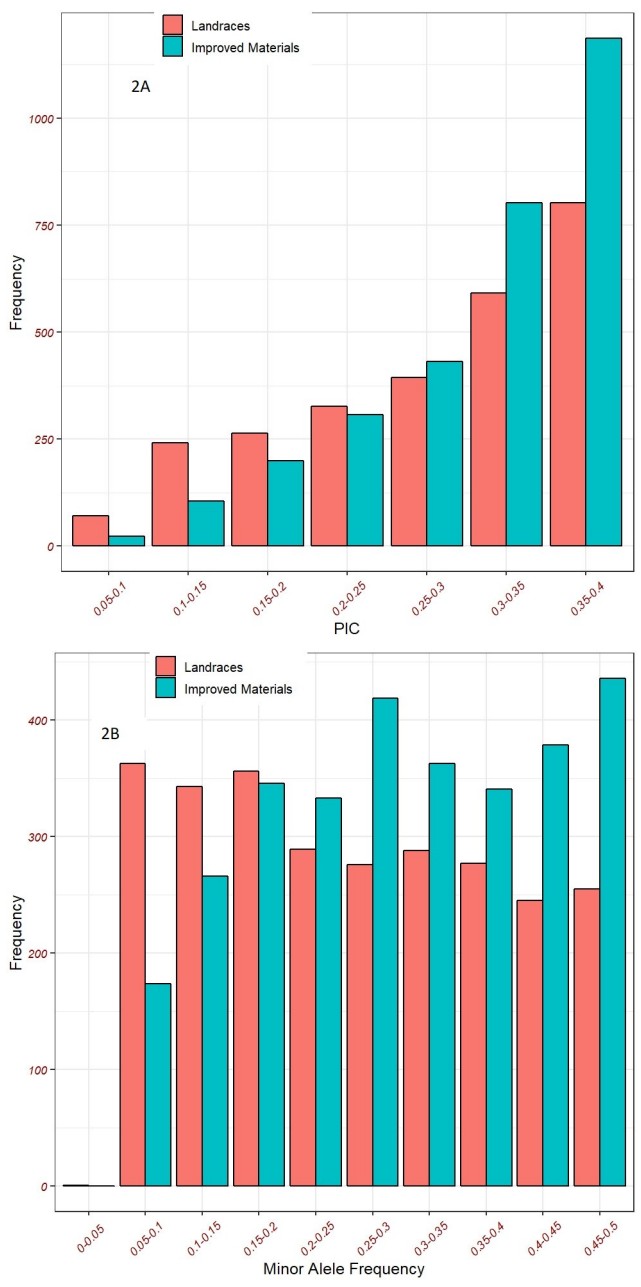

**Fig 2.** The distribution of polymorphism information content [PIC] [A] and allele frequency [B] for the improved accessions and landraces.

## Phenotypic evaluation

Field response scores across environments were normally distributed with 14.3% CV and heritability of 85%. Consequently, hereafter we will report the FR results only. The analysis of variance for the FR observations indicated a significant effect for the four environments and genotypes. However, no significant effect was detected for the genotypes × environments interaction (S1 Table). Based on the average of the FR across environments, the percentages of the resistant (R), moderately resistant (MR), moderately susceptible (MS), and susceptible (S)

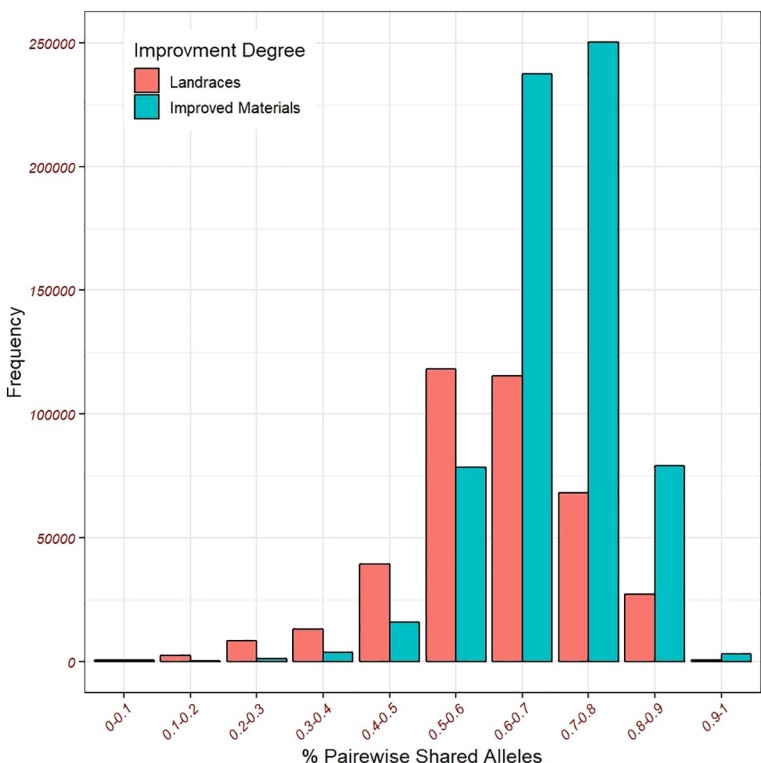

**Fig 3. The distribution of the pair-wise shared alleles for the improved accessions and the landraces.**

accessions in the landraces were 8, 27, 26, and 29%, respectively. The percentages of the field responses for the improved accessions were 24, 50, 16 and 10% for the R, MR, MS and S types, respectively. Across environments, the values of the field responses for the local check cultivars were 4.6, 3.75 and 3.37 for Gimmiza9, Giza168, and Sids13, respectively. Thus, the three check cultivars were considered moderately resistant. The overall mean of the resistance scores for the improved accessions and landraces was 4, and 5.25, respectively, indicating that improved accessions were more resistant to the stripe rust pathotypes present. Furthermore, from the improved accessions, 140 (13% of this class) genotypes found to be more resistant to stripe rust compared to the best check cultivar (Fig 4 and S1 Table). Also, 51 (5% of this class) landraces outperformed the best check cultivar in terms of resistance to stripe rust (Fig 4 and S2 Table).

## Marker-Trait association for stripe rust

Linkage disequilibrium (LD) was estimated for the improved accessions and landraces using 3215 SNP markers. LD declined to 50% of its initial values at 10 cM for the improved accessions and 11.75 cM for the landraces (Fig 5). Bayesian information criteria (BIC) was used to determine the optimum number of principal components (PCA) to use in order to account for the population stratification. BIC results indicated that the mixed model with no PCA was the optimum model to use. Furthermore, to validate the previous results, the percentage of variance that the first PCA accounted for was calculated, which was less than 1% of the total variance. Therefore, we reported the results of the association mapping using only the kinship (K) matrix which accounted for most of the stratification among genotypes in the landraces and improved accessions.

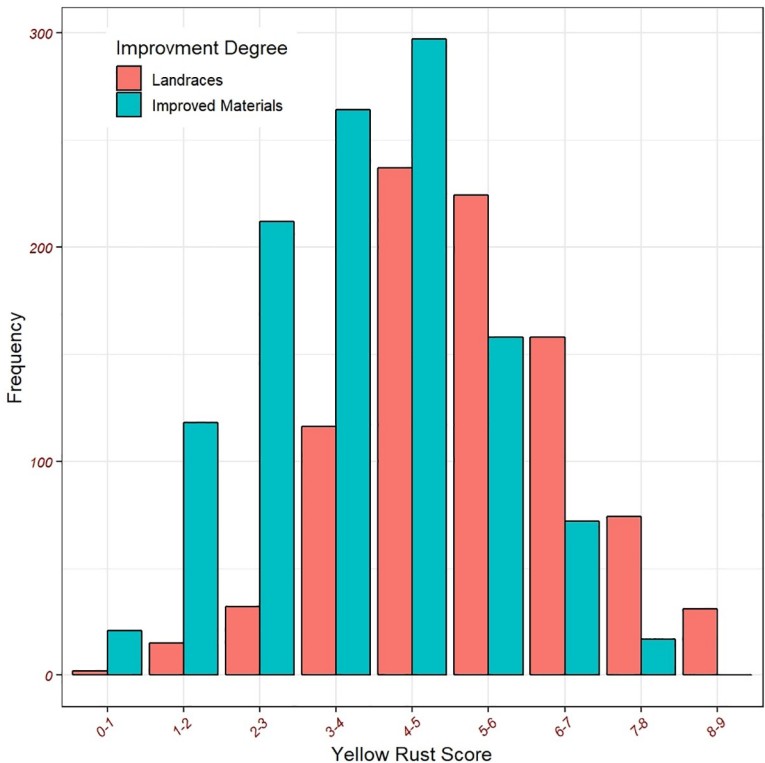

**Fig 4. The distribution of the yellow rust resistance score across environments for the improved accessions and landraces.**

Genotype × Environment interaction was not significant, thus we applied genome-wide association mapping analysis (GWAS) using the means across environments for the landraces and the improved accessions. Overall, GWAS identified 24 SNPs that were associated with stripe rust resistance (Table 1). Eight SNPs were common between landraces and improved accessions, while the rest were either significant only for the landraces (eight SNPs) or the improved accessions (another eight SNPs) (Fig 6). The common SNPs were located on chromosomes 1B (IWA4349 and IWA6787), 2B (IWA4349 and IWA6787), 5A (IWA6988), 6A (IWA5142), and 7B (IWA3415 and IWA3416). Of the eight SNPs that were found to be significantly linked with stripe rust in the landraces only, five of these eight SNPs were located in chromosome 1A (IWA6644, IWA3182, IWA5150, IWA4351 and IWA6649), and three in 1B (IWA5370, IWA7331, and IWA3892). The eight SNPs that were found to be significantly associated with stripe rust resistance in the improved accessions were located on chromosomes 1B (IWA7048 and IWA4155), 2B (IWA4096, IWA4095, IWA4097 and IWA7371), 5A (IWA7 880) and 5B (IWA3514) (Fig 6).

Among the significant SNPs, IWA7331 and IWA7048 had the lowest minor allele frequency of 0.07 and 0.09, respectively. The percentage of variance explained by the significant markers, in the landraces, ranged from 0.70 to 10.01%, while in the improved accessions, it ranged from 0.83 to 14.53% (Table 1). All significant markers in the landraces had a positive additive effect on the stripe rust resistance. While, in the improved accessions three of the significant markers in chromosome 2B (IWA4095, IWA4097 and IWA7371) had a negative additive effect.

The effect of the number of favorable alleles on the stripe rust resistance (Fig 7) indicated that as the number of favorable alleles increases, the stripe rust score decreased (evel of

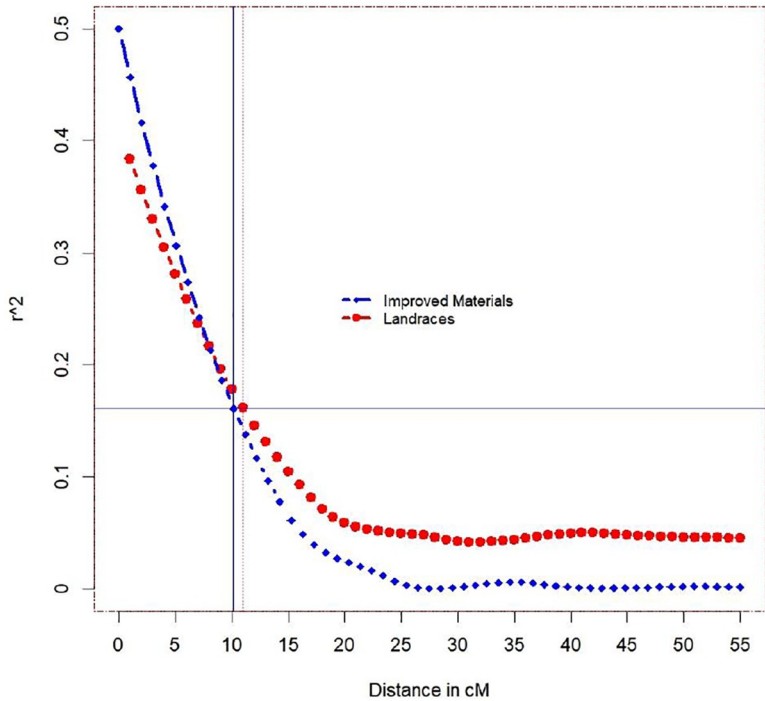

**Fig 5. Genome-wide linkage disequilibrium (LD) decays for the landraces and the improved accessions using 3215 SNP markers.**

resistance increased). Furthermore, there was a highly significant negative correlation between the number of favorable alleles in the landraces (r = - 0.65, P-value < 0.01)and improved accessions (r = -0.73, P-value < 0.01) with respect to stripe rust FR scores (Fig 7).

The results of the in silico analysis indicated that the 24 SNPs identified in our study are located in 18 wheat functional genes. Several of the significant SNPs identified were located in the same gene. For example, IWA4351 and IWA6649 are located in TraesCS1A01G015900 gene, While, IWA4349 and IWA6787 are located in TraesCS1B01G046300 gene. Furthermore, IWA4095, IWA4096, IWA4097 and IWA7371 SNPs are located in TraesCS2B01G501500 gene (Table 2).

## Discussion

Stripe rust is an airborne wheat disease with dynamic virulence evolution, thus anticipatory and continuous screening in the hotspot regions is crucial to identify and integrate new resistance resources and predict disease epidemics [42]. Plant breeders must remain vigilant as the stripe rust spores have the capacity for long-distance migration via airborne pathways [43,44]. The previous history and circumstances combined with the anticipated global warming might stimulate the presence of new stripe rust pathotypes, which creates an urgent need to develop new stripe rust resistant lines [45]. Therefore, identifying such lines in the stripe rust hotspot regions, i.e., Egypt, which currently experiences the presence of new pathotypes, can be beneficial to other breeders in other geographic regions that likely to face the same challenge in the future [46].

Despite the fact that the environments used in our study were different statistically, the magnitude of differences was rather small. Moreover, a highly significant statistical difference was detected among genotypes, but genotype × environment interaction was not significant

**Table 1. Significant markers associated with stripe rust resistance in the landraces and improved accessions.**

| Improvement Degree | SNP | chrom | Position (cM) | P.value | MAF | Additive effect | R² |
|---|---|---|---|---|---|---|---|
| Landraces | IWA6644 | 1A | 8.3 | 1.25E-07 | 0.48 | 0.29 | 4.50 |
| | IWA3182 | 1A | 9.5 | 3.58E-11 | 0.28 | 0.24 | 0.79 |
| | IWA5150 | 1A | 9.9 | 2.13E-18 | 0.23 | 0.30 | 10.01 |
| | IWA4351 | 1A | 11.6 | 7.4E-27 | 0.36 | 0.32 | 5.94 |
| | IWA6649 | 1A | 11.6 | 7.81E-26 | 0.36 | 0.26 | 6.19 |
| | IWA5370 | 1B | 11 | 3.66E-07 | 0.11 | 0.25 | 0.96 |
| | IWA7331 | 1B | 11 | 4.21E-08 | 0.07 | 0.23 | 0.70 |
| | **IWA4349** | 1B | 13.2 | 8.73E-16 | 0.33 | 0.39 | 8.04 |
| | **IWA6787** | 1B | 13.2 | 6.41E-13 | 0.35 | 0.21 | 7.05 |
| | IWA3892 | 1B | 123.4 | 1.61E-06 | 0.37 | 0.36 | 0.90 |
| | **IWA7799** | 2B | 46.9 | 5.59E-12 | 0.23 | 0.23 | 1.37 |
| | **IWA6121** | 2B | 206.2 | 4.65E-06 | 0.20 | 0.38 | 2.70 |
| | **IWA6988** | 5A | 190.4 | 1.93E-07 | 0.40 | 0.39 | 6.66 |
| | **IWA5142** | 6A | 131.8 | 2.43E-06 | 0.35 | 0.30 | 2.83 |
| | **IWA3415** | 7B | 164.9 | 5.73E-12 | 0.19 | 0.50 | 7.99 |
| | **IWA3416** | 7B | 164.9 | 4.22E-12 | 0.19 | 0.17 | 7.83 |
| Improved Accessions | **IWA4349** | 1B | 13.2 | 2.7E-22 | 0.17 | 0.56 | 10.61 |
| | **IWA6787** | 1B | 13.2 | 1.93E-18 | 0.23 | 0.18 | 9.04 |
| | IWA7048 | 1B | 22.9 | 5.7E-08 | 0.09 | 0.35 | 0.83 |
| | IWA4155 | 1B | 96.4 | 5E-15 | 0.49 | 0.37 | 2.41 |
| | **IWA7799** | 2B | 46.9 | 2.74E-19 | 0.43 | 0.39 | 2.61 |
| | IWA4096 | 2B | 199.3 | 2.78E-08 | 0.28 | 0.39 | 2.43 |
| | IWA4095 | 2B | 200.5 | 3.46E-07 | 0.26 | -1.18 | 1.95 |
| | IWA4097 | 2B | 200.5 | 1.88E-06 | 0.26 | -2.10 | 1.89 |
| | IWA7371 | 2B | 200.5 | 1.36E-06 | 0.26 | -0.89 | 1.90 |
| | **IWA6121** | 2B | 206.2 | 3.21E-09 | 0.33 | 0.31 | 1.89 |
| | **IWA6988** | 5A | 190.4 | 1.17E-11 | 0.15 | 0.35 | 2.66 |
| | IWA7880 | 5A | 190.4 | 3.92E-14 | 0.23 | 0.38 | 2.53 |
| | IWA3514 | 5B | 22.9 | 4.24E-17 | 0.36 | 0.39 | 2.44 |
| | **IWA5142** | 6A | 131.8 | 6.62E-09 | 0.45 | 0.39 | 3.86 |
| | **IWA3415** | 7B | 164.9 | 9.56E-27 | 0.39 | 0.48 | 14.55 |
| | **IWA3416** | 7B | 164.9 | 4.04E-27 | 0.39 | 0.27 | 14.53 |

$R^2$: Percentage of the variance explained. **Bold SNP names** refer to significant markers in the landraces and the improved accessions. **MAF:** minor allele frequency

indicating the environments were similar in their pathogenicity. These results may be due to successfully inoculating the accessions with the same five pathotypes across environments. Tsilo et al. (2014) [47] reported a similar effect of the artificial inoculation with a known mixture of leaf rust pathotypes in five environments, in which accessions in the artificially inoculated environments tended to have similar effects across environments. Furthermore, the surrounding farms around the experimental locations during the two growing seasons were commercially cultivated with the same wheat cultivars, i.e, Gimmiza9 and Sids12, which were officially the recommended cultivars for both locations. Inoculating with the same stripe rust pathotypes, in addition to having the same surrounding cultivars, might have created a similar environmental effect on the studied accessions [48]. These results help to explain the observed high broad sense heritability (85%) which also indicates that most of the variance observed in the current study can be attributed to differences among the studied accessions [49].

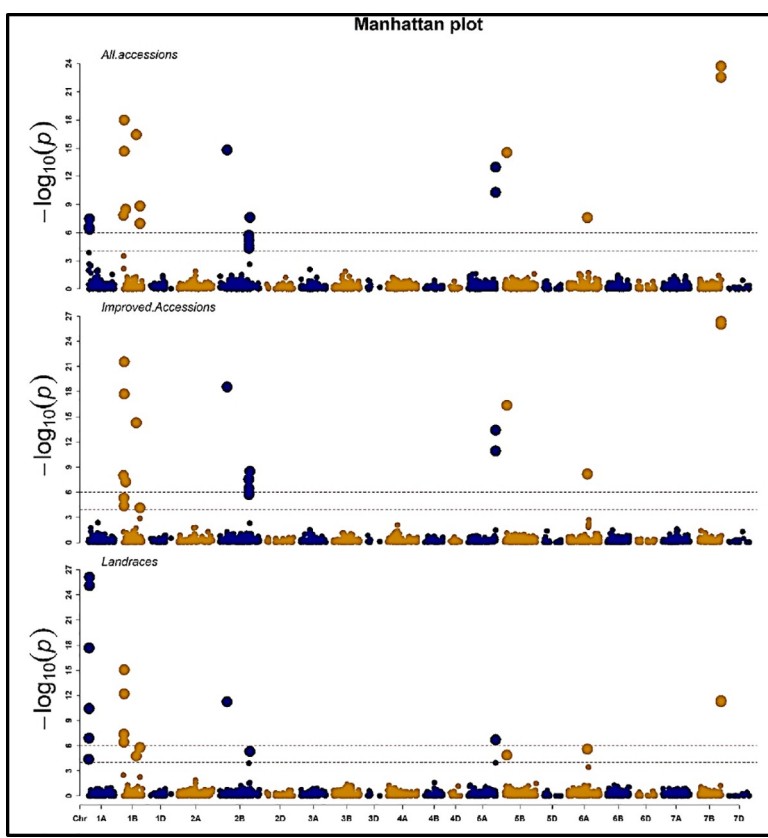

**Fig 6. Manhattan** plot for stripe rust results obtained from genome-wide association mapping for all accessions (A), improved accessions (B) and landraces (C).

Investigating the difference among accessions indicated that landraces tend to have more minor alleles and fewer pairwise shared alleles compared to the improved accessions. One explanation for this result is the active germplasm exchange among the spring wheat breeding programs. Also, selection for traits such as plant height and grain yield might also have resulted in a change in the frequency of other linked alleles [50]. Additionally, the overall mean of stripe rust score across environments in the landraces was 31% higher than that in the improved accessions, indicating that breeding efforts during the last century have resulted in improved stripe rust resistance. Nevertheless, 42 landraces were resistant to stripe rust and outperformed the resistant check cultivars indicating that excellent levels of resistance in the landraces have evolved. The resistant landraces through the natural selection might contain novel resistance genes or combinations of resistance gene that would be valuable for stripe rust breeding efforts [51]. Overall, our results indicated that the landraces and improved accessions are structurally different. That structural difference, caused by differences in minor allele frequency and pairwise alleles sharing, can be beneficial or detrimental. It can be useful in identifying complementary genomic regions or genes to improve resistance, but it can also increase the false discovery rate in the genome-wide association mapping (GWAS) studies if the landraces and improved accessions were fitted simultaneously to the same model [4].

Consequently, separate GWAS models were fitted for the landraces and the improved accessions. In the same context, linkage disequilibrium (LD) measured in the improved accessions tended to decay more rapidly than that measured in the landraces. The causes for the fast decay in the improved accessions is due to recombination (most likely resulting from

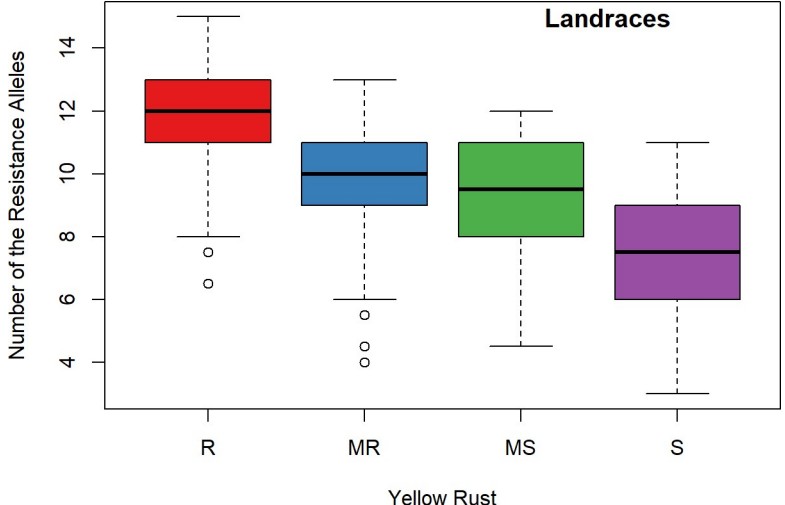

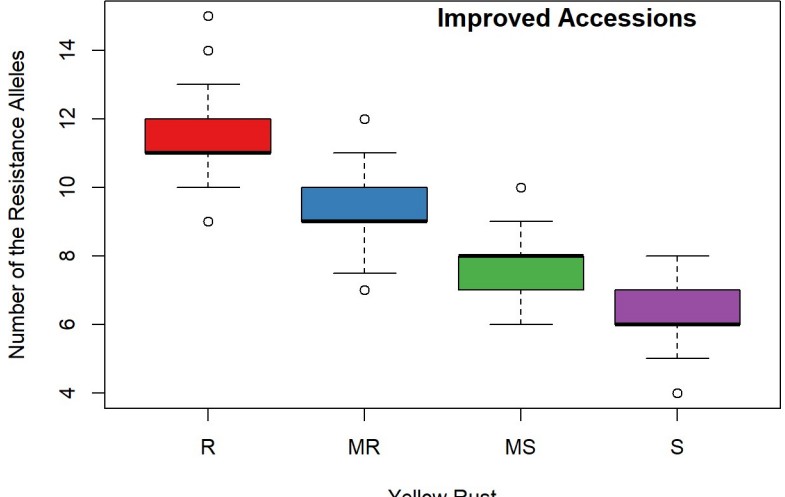

**Fig 7. Boxplot for the number of favorable alleles effect on stripe rust resistance for the improved accessions and the landraces.**

breeders'designed crosses using shared germplasm) followed by selection for particular alleles [52]. The LD decay with genetic distance observed our study indicates that GWAS is a useful approach to identifying SNPs linked to stripe rust resistance genes in the landraces and the improved accessions. GWAS results indicated that eight SNPs were significantly linked with stripe rust resistance exclusively in the landraces. Five of these eight SNPs were in chromosome 1A in the region from 8.3 to 11.6 cM. This region in the 1A chromosome contains two QTLs, i.e., QYrid.ui-1A_RioBlanco [15] and QYr.tam-1A_Avocet-YrA [53]. Another two SNPs [IWA5370 and IWA7331] which were identified in previous studies and found to be linked with stripe rust resistance genes might represent *Yr3a*, *Yr3b*, *Yr3c*, or *Yr21* [19]. IWA3892 is another SNP marker found to be associated with the stripe rust resistance gene in the landraces [15].

**Table 2. Represents genes that have significant stripe rust-resistant markers.**

| SNP | Allele | Gene ID | Function |
|---|---|---|---|
| IWA3182 | A/G | TraesCS1A01G012500 | Transforming growth factor-beta receptor-associated protein 1 |
| IWA3415 | A/G | TraesCS7B01G476800 | Calcium-binding EF hand protein-like |
| IWA3416 | T/C | TraesCS7B01G476800 | Calcium-binding EF hand protein-like |
| IWA3514 | T/C | TraesCS5B01G019600 | Amino acid permease family protein, putative, expressed |
| IWA3892 | A/G | TraesCS1B01G449600 | Auxilin-like protein 1 |
| IWA4095 | A/G | TraesCS2B01G501500 | UHRF1-binding protein 1 |
| IWA4096 | T/C | TraesCS2B01G501500 | UHRF1-binding protein 1 |
| IWA4097 | A/G | TraesCS2B01G501500 | UHRF1-binding protein 1 |
| IWA4155 | A/G | TraesCS1B01G396200 | 1,2-dihydroxy-3-keto-5-methylthiopentene dioxygenase |
| IWA4349 | T/C | TraesCS1B01G046300 | ACT domain-containing protein |
| IWA4351 | A/G | TraesCS1A01G015900 | Serine/threonine-protein kinase |
| IWA5142 | T/C | TraesCS6A01G348400 | Amino acid transporter family protein |
| IWA5150 | T/G | TraesCS1A01G015200 | Tubulin-specific chaperone cofactor E-like protein |
| IWA5370 | T/G | TraesCS1B01G030800 | ABC transporter ATP-binding protein |
| IWA6121 | A/G | TraesCS2B01G495700 | RNA-binding family protein |
| IWA6644 | T/C | TraesCS1A01G005800 | Mei2-like protein |
| IWA6649 | T/C | TraesCS1A01G015900 | Serine/threonine-protein kinase |
| IWA7048 | T/C | TraesCS1B01G069300 | DUF810 family protein |
| IWA7331 | T/G | TraesCS1B01G041300 | Transducin/WD40 repeat protein |
| IWA7371 | T/C | TraesCS2B01G501500 | UHRF1-binding protein 1 |
| IWA7799 | T/G | TraesCS2B01G074100 | basic helix-loop-helix [bHLH] DNA-binding superfamily protein |
| IWA7880 | T/G | TraesCS5A01G537100 | Nitrate transporter 1.1 |

Eight SNP markers (IWA7048, IWA4155, IWA4096, IWA4095, IWA4097, IWA7371, IWA7880 and IWA3514) were significantly associated with stripe rust resistance genes in the improved accessions, but not in the landraces. IWA7048 was previously associated with grain total protein content [54]. However, no published reports were found to associate IWA4155 SNP marker with stripe rust resistance genes. Recently, IWA4096, IWA4095 and IWA4097 were found to be linked with stripe rust resistance genes in Ethiopian durum wheat (*Triticum turgidum ssp. durum)* [55]. IWA7371 is located in the same genomic region that contains *Yr5* gene; however, no previous reports have identified an association between that marker and *Yr5*gene. Therefore, IWA7371 might be a novel marker for that gene or an unknown gene. Additionally, IWA7880 and IWA3514 SNP markers located in chromosomes 5A and 5B, respectively, had not previously been associated with stripe rust resistance genes.

An additional eight SNPs (IWA4349, IWA6787, IWA7799, IWA6121, IWA6988, IWA5142, IWA3415 and IWA3416) were significantly linked with stripe rust resistance genes in both the landraces and the improved accessions. Out of these eight markers, IWA4349 [47] and IWA6988 [15] were previously associated with stripe rust resistance genes. IWA6121 was found to be tightly linked to *Yr5* [56]. IWA3415 and IWA3416 were found to be associated with stripe rust resistance gene in spring wheat [15]. Furthermore, three of the markers were found to be linked with stripe rust in the landraces and the improved accessions (IWA6787, IWA7799, and IWA5142) were not reported before. Therefore, most likely these SNPs are novel markers for stripe rust resistance.

Functional gene annotation for the significant 24 SNPs indicated that these SNPs are located in 18 wheat functional genes. Four of these 18 genes have known products that contribute directly or indirectly in several plant defense mechanisms. For example, receptor-like

kinases (gene: TraesCS1A01G015900, SNP: IWA4351 and IWA6649) play a crucial role in the plant ability to recognize both general elicitors and specific pathogens through resistance (R) gene [57]. ACT domain-containing protein (gene: TraesCS1B01G046300, SNP: IWA4349 and IWA6787) serve as amino acid-binding sites in several feedback-regulated amino acid metabolic enzymes [58]. Calcium-binding EF-hand protein-like (gene: TraesCS7B01G476800, SNP: IWA3415 and IWA3416) involved in transmembrane signal transductions and is important for plant disease resistance [59]. UHRF1-binding protein1 (gene: TraesCS2B01G501500, SNP: IWA4095, IWA4096, IWA4097 and IWA7371) plays a critical role in DNA methylation and is a regulator of cell proliferation [60]. The other 14 genes have no known or published information about their contribution to the plant defense mechanisms.

Identification of favorable alleles for stripe rust resistance is a prerequisite to enhancing the resistance of the modern cultivars by introgression and accumulating several favorable alleles from the wheat gene pool through molecular markers. In this study, the correlation between several favorable alleles combinations and stripe rust resistance was highly significant and biologically meaningful. Therefore, these favorable alleles of stripe rust resistance would be useful for understanding and improving wheat stripe rust resistance.

## Conclusion

This study is one of the first large-scale studies to be conducted in the Mediterranean basin and in one of the global stripe rust hotspot regions. Generating new stripe rust phenotypic information on the studied panel while using the publicly available molecular marker data, contributed to identifying potentially novel QTLs associated with stripe rust for this region and validated 17 of the previously reported QTLs under the field conditions.

Overall, the improved accessions tended to be more resistant to stripe rust compared to the landraces. Out of the 24 QTLs that were found to be significantly associated with stripe rust, 17 were previously reported, while seven are potentially novel. Stripe rust resistant accessions identified in the current study will be included in various crossing blocks to enhance stripe rust resistance in Egyptian elite lines. These and previous findings will contribute to plant breeders' and pathologists' efforts in Egypt, North Africa, and the Mediterranean basin to improve the overall resistance to stripe rust by providing new resources and highlighting the importance of using markers to improve selection accuracy.

## Supporting information

**S1 Table. Analysis of variance for field responses of stripe rust across environments.** (XLSX)

**S2 Table. List of stripe rust resistant accessions, along with their improvement degree and country of origin.** (XLSX)

**S1 Fig.** Heatmap estimated from the pairwise shared alleles for the improved accessions [A] and the landraces [B]. (TIF)

## Acknowledgments

Without the wheat germplasm that was collected by the Triticeae Coordinated Agricultural Project (TCAP), the reported research will not be possible. Thus, we sincerely appreciate the great work of TCAP team. We also would like to thank the Arab fund fellowship program for

supporting this work. We thank both Mr. Hafiz Mazeek and Mr. Talaat Salah, Damanhur University, Egypt, for their effort during this experiment. We also would like to show our gratitude to the graduate and undergraduate students who helped us during the summer fieldwork.

## Author Contributions

**Conceptualization:** Ibrahim S. Elbasyoni, Walid M. El-Orabey, Sabah Morsy, P. S. Baenziger, Ismail Dowikat.

**Data curation:** Ibrahim S. Elbasyoni, Walid M. El-Orabey, Sabah Morsy.

**Formal analysis:** Ibrahim S. Elbasyoni, P. S. Baenziger.

**Funding acquisition:** Ibrahim S. Elbasyoni, Sabah Morsy, P. S. Baenziger.

**Investigation:** Ibrahim S. Elbasyoni, Sabah Morsy.

**Methodology:** Ibrahim S. Elbasyoni, Walid M. El-Orabey, Sabah Morsy, Ismail Dowikat.

**Project administration:** Ibrahim S. Elbasyoni.

**Resources:** Ibrahim S. Elbasyoni, Sabah Morsy.

**Supervision:** Ibrahim S. Elbasyoni, Sabah Morsy.

**Validation:** Ibrahim S. Elbasyoni, Sabah Morsy, P. S. Baenziger.

**Visualization:** Ibrahim S. Elbasyoni, Sabah Morsy, P. S. Baenziger.

**Writing – original draft:** Ibrahim S. Elbasyoni, Walid M. El-Orabey, Sabah Morsy, P. S. Baenziger, Zakaria Al Ajlouni, Ismail Dowikat.

**Writing – review & editing:** Ibrahim S. Elbasyoni, Walid M. El-Orabey, P. S. Baenziger, Zakaria Al Ajlouni, Ismail Dowikat.

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
