## [Decision Letter · Decision Letter 0]

28 Jun 2019

PONE-D-19-14766

Evaluation of a Global Spring Wheat Panel for Stripe Rust: Resistance Loci Validation and Novel Resources Identification

PLOS ONE

Dear Dr Elbasyoni,

Thank you for submitting your manuscript to PLOS ONE. After careful consideration, we feel that it has merit but does not fully meet PLOS ONE’s publication criteria as it currently stands. Therefore, we invite you to submit a revised version of the manuscript that addresses the points raised during the review process.

Dear Dr. Elbasyoni,

I received comments from the advisers on your manuscript "Evaluation of a Global Spring Wheat Panel for Stripe Rust: Resistance Loci Validation and Novel Resources Identification" which you submitted to PlosONE. Based on reviewer comments and my assessment I have decided that your manuscript could be reconsidered for publication should you be prepared to incorporate major revisions. When preparing revised manuscript, you are asked to carefully consider the reviewer comments, which can be found below, and submit a list of detailed and itemized responses to the comments. Please take attention to comment of reviewer one regarding use of an average data.

With kind regards,

Dragan

We would appreciate receiving your revised manuscript by Aug 12 2019 11:59PM. To enhance the reproducibility of your results, we recommend that if applicable you deposit your laboratory protocols in protocols.io, where a protocol can be assigned its own identifier (DOI) such that it can be cited independently in the future. For instructions see: http://journals.plos.org/plosone/s/submission-guidelines#loc-laboratory-protocols

We look forward to receiving your revised manuscript.

Kind regards,

Dragan Perovic, Ph.D

Academic Editor

PLOS ONE

Journal Requirements:

Additional Editor Comments:

Dear Dr. Elbasyoni,

I received comments from the advisers on your manuscript "Evaluation of a Global Spring Wheat Panel for Stripe Rust: Resistance Loci Validation and Novel Resources Identification" which you submitted to PlosONE. Based on reviewer comments and my assessment I have decided that your manuscript could be reconsidered for publication should you be prepared to incorporate major revisions. When preparing revised manuscript, you are asked to carefully consider the reviewer comments, which can be found below, and submit a list of detailed and itemized responses to the comments. Please take attention to comment of reviewer one regarding use of an average data.

With kind regards,

Dragan

Reviewers' comments:

Reviewer's Responses to Questions

**Comments to the Author**

1. Is the manuscript technically sound, and do the data support the conclusions?

Reviewer #1: Partly

Reviewer #2: Partly

2. Has the statistical analysis been performed appropriately and rigorously? 

Reviewer #1: Yes

Reviewer #2: Yes

3. Have the authors made all data underlying the findings in their manuscript fully available?

Reviewer #1: Yes

Reviewer #2: Yes

4. Is the manuscript presented in an intelligible fashion and written in standard English?

Reviewer #1: Yes

Reviewer #2: Yes

5. Review Comments to the Author

Reviewer #1: Comments

Major comments

1) Combined data analysis: The authors presented the disease score result as an average of disease scores across environments and over time. Since these are field experiments, there may be natural infections (the manuscript presented strong arguments showing natural infections in the experimental locations: lines 118-128, 470-472 and 569-570). The response of accessions may vary depending on the prevalent race in each location and year. Hence, calculating averages may not provide an accurate picture of the reaction of accessions in each location and year. For example, an accession with resistant reaction in one location (R = 2) but susceptible reaction in the second location (S = 8) will have an average disease score value of 5, which may be categorized as MR = 4 or MS = 6. Both MR and MS are not actually the disease reactions for that specific accession. The result has to be presented as “the accession had resistant reaction in location 1 but susceptible reaction in location 2”. The QTL analysis should also be done for each location and year separately (the authors may refer an article in Plant Genome doi:10.3835/plantgenome2016.10.0107).

2) Selection of resistant accessions: The authors presented in Table S2 (supporting information) accessions that were selected as resistant. They picked only accessions that were having resistant reaction (disease score of 3 and less). This will exclude accessions that were moderately resistant. An accession with MR disease reaction to yellow rust is considered as a good source of stripe rust resistance, hence have to be included in the selection. Sometimes depending on the disease severity accessions with MS reaction could also be a good source of adult plant resistance.

Minor comments

- Shorten the introduction

Line 140: …. Stripe rust resistance

Line 170: delete ‘was’

Line 181: Field disease reaction of germplasms to rusts (R, MR, MS, S) at adult plant stage is not referred as infection type (IT). IT is a term used to refer reaction to seedling reaction. The words for adult plant response are ‘Field Response’. Hence, change IT to field response throughout the manuscript.

Line 344: What does ‘infection value’ indicate? Use terms consistently across the manuscript.

Lines 344-346: What was the cut point to distinguish R from MR, MR from MS and MS from S? This has to be indicated in the materials and methods part. As it stands, Gimiza9 with value of 4.6 is closer to MR = 4 than MS = 6.

Line 347: what does resistance score represent? Be consistent

Line 466: delete ‘that’

Line 477: environments

Table 1. R2 values are between 0 – 1. Why are there values greater than 1?

TableS2. Field response of the selected genotypes in both years and location should be presented separately, i. e., Elbostan-2016, Elbostan-2017, Elkhazan-2016 and Elkhazan-2017.

Reviewer #2: Dear Dr. Ibrahim Elbasyoni,

the manuscript deals with stripe rust as one of the most important fungal wheat diseases. Occurence of new races has been observed many times in the past. Warrior races led in Europe to massive infections. Hence it is of high interest to evaluate genetic ressources which can increase the resistance of cultivars in the future. Therefore,as one result, shown in the manuscript, resistant genotypes can be selected for breeding, supported by SNP markers, which are easy to handle. However some aspects can improve the manuscript:

Please check the English spelling in some parts. At the beginning you use "Identifying" several times. Perhaps, some alternatives should be integrated into the text.

Other points:

Line:

64 to 66: Check sentence (tense).

69: change letters "Yr" to Italic style

entire Introduction:

PLease change both mentioned Yr.. to Italic style

Line 127:

Please include a citation into the sentence about Warrior races

Line 155: I would like to suggest to change from "an experimental farm for" to

"field station of"

Line 170: Divide into two words

Line 170 to 173: Please show approximately the spore amounts (e.g. 20 mg per

slot)... How was the germination rate of spores (if data are available). At which

time of the day, inoculation was performed.

Line 203: Please cite original source of wnsp 2013 consensus map

Line 390 (Table 1: Please indicate in the description of the Table,that R²

describes the percentage of explained phenotypic variance.

Line 418: change frequenc to frequency

Line 453: No information about the method to get the data for Table 2 were

provided in the Material and Method (MM) section. Is this a blast of markers to

genes in a database? Did you use the reference genome to check this informations?

Please insert method into MM section. Are these genes the best hits or did you

select genes, specific for defence reactions?

Table S1, Please show the high heritability of the trait in the results already.

This heritability is very high in comparison to other studies (as you discussed).

Please check Table S2. Why only breeding lines are provided, all are Type R

independend from the Yr score. Please show in the head of the table, what Yr

score means. All should be self explaining.

6. PLOS authors have the option to publish the peer review history of their article (what does this mean?). If published, this will include your full peer review and any attached files.

Reviewer #1: Yes: Belayneh Admassu Yimer

Reviewer #2: No

---

## [Author Response · Author response to Decision Letter 0]

12 Jul 2019

Dear Reviewer#1:

Thank you so much for making time to read our manuscript. All your valuable comments, edits and suggestions are highly appreciated and we will do our best to make the best use of it. 

Below is our response detailed response to your comments:

#==================================================================================#

A- Major comments:

1) Combined data analysis: The authors presented the disease score result as an average of disease scores across environments and over time. Since these are field experiments, there may be natural infections (the manuscript presented strong arguments showing natural infections in the experimental locations: lines 118-128, 470-472 and 569-570). The response of accessions may vary depending on the prevalent race in each location and year. Hence, calculating averages may not provide an accurate picture of the reaction of accessions in each location and year. For example, an accession with resistant reaction in one location (R = 2) but susceptible reaction in the second location (S = 8) will have an average disease score value of 5, which may be categorized as MR = 4 or MS = 6. Both MR and MS are not actually the disease reactions for that specific accession. The result has to be presented as “the accession had resistant reaction in location 1 but susceptible reaction in location 2”. The QTL analysis should also be done for each location and year separately (the authors may refer an article in Plant Genome doi:10.3835/plantgenome2016.10.0107).

Thank you so much for your valuable comment and suggestion. We totally understand the merits of analyzing each location and year separately. However, as mentioned in the manuscript lines 335 and 336 all environments (locations and growing seasons were homogeneous) that means the lines grown in each location had the same pattern of response. Additionally, as shown in the text (lines 339 - 340) and table S1 no significant G*E was detected which also indicated that the lines had the same pattern of response across environments. In the current study, we were interested in identifying lines resistant to yellow rust across the studied environments that is why we are using that large number of accessions. As we also mentioned in the discussion, the studied accessions were surrounded by the same commercial cultivars across years. Therefore, we had no statistical or biological reason to conduct the analysis for each environment separately. 

2) Selection of resistant accessions: The authors presented in Table S2 (supporting information) accessions that were selected as resistant. They picked only accessions that were having resistant reaction (disease score of 3 and less). This will exclude accessions that were moderately resistant. An accession with MR disease reaction to yellow rust is considered as a good source of stripe rust resistance, hence have to be included in the selection. Sometimes depending on the disease severity accessions with MS reaction could also be a good source of adult plant resistance.

We totally understand your point that MR and MS could contain resistant lines. We did not ignore MR and MS lines. As a mater of fact all data is already in the depository for others to use. We simply presented, in the supporting information, the most resistant lines, that does not mean we ignored MS and MR lines, we already described that in the text. However, the uncertainty attached to using MR and MS genotypes is high and mostly reevaluating these lines will result in resistance to specific race or races or susceptibility. Meanwhile, in the current study, we were able to identify 330 resistant wheat genotypes that found to be resistant across environments, that genotypes are being re-evaluated in other locations for stripe rust.

B-Minor comments

- Shorten the introduction:

Introduction was shortened 

Line 140: …. Stripe rust resistance

The word resistance was added

Line 170: delete ‘was’

Deleted

Line 181: Field disease reaction of germplasms to rusts (R, MR, MS, S) at adult plant stage is not referred as infection type (IT). IT is a term used to refer reaction to seedling reaction. The words for adult plant response are ‘Field Response’. Hence, change IT to field response throughout the manuscript.

Thank you so much for the suggestion, IT was replaced by Field response throughout the manuscript. 

Line 344: What does ‘infection value’ indicate? Use terms consistently across the manuscript.

 Corrected accordingly 

Lines 344-346: What was the cut point to distinguish R from MR, MR from MS and MS from S? This has to be indicated in the materials and methods part. As it stands, Gimiza9 with value of 4.6 is closer to MR = 4 than MS = 6.

The cut points are presented in the materials and methods lines: 182:184 …. That is precisely why we use the average across locations…. 

Line 347: what does resistance score represent? Be consistent

Corrected accordingly 

Line 466: delete ‘that’

Deleted

Line 477: environments

Corrected 

Table 1. R2 values are between 0 – 1. Why are there values greater than 1?

That is because we are present it as the percentage of the variance explained, Footnote was added under the table. 

TableS2. Field response of the selected genotypes in both years and location should be presented separately, i. e., Elbostan-2016, Elbostan-2017, Elkhazan-2016 and Elkhazan-2017.

Please see our response to your first comment. 

Dear Reviewer#2:

Thank you so much for making time to read our manuscript. All your valuable comments, edits and suggestions are highly appreciated and we will do our best to make the best use of it. 

Below is our response detailed response to your comments:

Response to reviwer#2:

Reviewer #2: Dear Dr. Ibrahim Elbasyoni,

the manuscript deals with stripe rust as one of the most important fungal wheat diseases. Occurence of new races has been observed many times in the past. Warrior races led in Europe to massive infections. Hence it is of high interest to evaluate genetic ressources which can increase the resistance of cultivars in the future. Therefore,as one result, shown in the manuscript, resistant genotypes can be selected for breeding, supported by SNP markers, which are easy to handle. However some aspects can improve the manuscript:

Please check the English spelling in some parts. At the beginning you use "Identifying" several times. Perhaps, some alternatives should be integrated into the text.

Synonyms to the word identifying ware used 

Spelling check was conducted 

Other points:

Line:

64 to 66: Check sentence (tense).

Sentence checked 

69: change letters "Yr" to Italic style

entire Introduction:

PLease change both mentioned Yr.. to Italic style

When we refer to gene names as italic, while when referring to yellow rust as a disease we used the regular style. 

Line 127:

Please include a citation into the sentence about Warrior races

References were inserted 

Line 155: I would like to suggest to change from "an experimental farm for" to

"field station of"

Thank you for the suggestion, but we are required to use “experimental farm” in my institution. 

Line 170: Divide into two words

The suggested correction was conducted 

Line 170 to 173: Please show approximately the spore amounts (e.g. 20 mg per

slot)... How was the germination rate of spores (if data are available). At which

time of the day, inoculation was performed.

The total weight of stripe rust spores that were used in inoculation was 200 mg per environment. Dusting was carried out in the early evening (at sunset) before dew formation. There is no data available about the rate of spore germination.

Line 203: Please cite original source of wnsp 2013 consensus map

Reference for the “wnsp2013” was inserted 

Line 390 (Table 1: Please indicate in the description of the Table,that R²

describes the percentage of explained phenotypic variance.

R2 description was included 

Line 418: change frequenc to frequency

Suggested correction was conducted 

Line 453: No information about the method to get the data for Table 2 were

provided in the Material and Method (MM) section. Did you use the reference genome to check this informations?

Please insert method into MM section.

 Description for the gene identification was added

Is this a blast of markers to genes in a database?

Yes, we used the the Triticeae Toolbox database to search for gene identification and functional annotation using IWGSC RefSeqv1.0

Are these genes the best hits or did you select genes, specific for defence reactions?

Yes, genes selected were the best hits 

Table S1, Please show the high heritability of the trait in the results already.

We did not understand the meaning of this point? If you want us to include the heritability values in the table, we already talked about it in the text. 

This heritability is very high in comparison to other studies (as you discussed).

As you mentioned we provided an explanation for the high broad sense heritability observed in our study, please see lines from 521 to 535 in the discussion 

Please check Table S2. Why only breeding lines are provided, all are Type R independend from the Yr score. Please show in the head of the table, what Yr score means. All should be self explaining.

In the header of the table we added: “yellow rust score and field response”

IT was replaced by field response as suggested by reviwer#1, also “type” was replaced by “field response”.

TablS2: contains 122 breeding lines , 111 cultivar and 97 landraces. 

Furthermore, in table S2 we are presenting only the resistant accessions across all environments, please see materials and methods [ lines 186:198].

---

## [Decision Letter · Decision Letter 1]

27 Aug 2019

PONE-D-19-14766R1

Evaluation of a Global Spring Wheat Panel for Stripe Rust: Resistance Loci Validation and Novel Resources Identification

PLOS ONE

Dear Dr Elbasyoni,

Thank you for submitting your manuscript to PLOS ONE. After careful consideration, we feel that it has merit but does not fully meet PLOS ONE’s publication criteria as it currently stands. Therefore, we invite you to submit a revised version of the manuscript that addresses the points raised during the review process.

We would appreciate receiving your revised manuscript by Oct 11 2019 11:59PM. To enhance the reproducibility of your results, we recommend that if applicable you deposit your laboratory protocols in protocols.io, where a protocol can be assigned its own identifier (DOI) such that it can be cited independently in the future. For instructions see: http://journals.plos.org/plosone/s/submission-guidelines#loc-laboratory-protocols

We look forward to receiving your revised manuscript.

Kind regards,

Dragan Perovic, Ph.D

Academic Editor

PLOS ONE

Additional Editor Comments (if provided):

Dear Dr Elbasyoni,

according to reports of our reviewers you manuscript could be considered for publication if you are willing to incorporate minor changes.

Hope that your respond will be quick in order to avoid further delays.

Regards

Dragan

Reviewers' comments:

Reviewer's Responses to Questions

**Comments to the Author**

1. If the authors have adequately addressed your comments raised in a previous round of review and you feel that this manuscript is now acceptable for publication, you may indicate that here to bypass the “Comments to the Author” section, enter your conflict of interest statement in the “Confidential to Editor” section, and submit your "Accept" recommendation.

Reviewer #1: (No Response)

Reviewer #2: All comments have been addressed

2. Is the manuscript technically sound, and do the data support the conclusions?

Reviewer #1: Yes

Reviewer #2: Yes

3. Has the statistical analysis been performed appropriately and rigorously? 

Reviewer #1: Yes

Reviewer #2: Yes

4. Have the authors made all data underlying the findings in their manuscript fully available?

Reviewer #1: Yes

Reviewer #2: Yes

5. Is the manuscript presented in an intelligible fashion and written in standard English?

Reviewer #1: Yes

Reviewer #2: Yes

6. Review Comments to the Author

Reviewer #1: 1) The cut points described in lines 196 - 198 are for single observations (R = 2, MR = 4, MS = 6 and S = 8). They don't take in to account average values. What happens if the average value is 5? Will it be MR or MS? That is why you need to describe your cut off point. On the same lines (355 - 358) you described Gimiza9 with average value of 4.6 as MS. I don't believe it's MS, rather it is closer to MR than MS. Describing the cut off value will avoid such confusions.

2) Lines 201 - 202: The authors described how they calculated the ACI; but at no point did they use the ACI values in the results or discussions part. Hence, you may delete the description in the materials and methods part.

3) The introduction part is still too long. It can be significantly shortened.

Reviewer #2: Dear Dr Ibrahim Elbasyoni,

all comments have been addressed. Critical comments have been considered. Please change only line 107: "accession" to accessions. The manuscript provides now valuable information's about genotypes, molecular markers, usable for resistance breeding against yellow rust. The northern regions of Africa are effected by many abiotic and biotic stresses. Yellow rust epidemics have been observed in European countries as well, so that genotypes carrying effective resistances in this study are usable in the future as the basis for an increased resistance level of elite material.

7. PLOS authors have the option to publish the peer review history of their article (what does this mean?). If published, this will include your full peer review and any attached files.

Reviewer #1: Yes: Belayneh Admassu Yimer

Reviewer #2: Yes: Dr. Albrecht Serfling

---

## [Author Response · Author response to Decision Letter 1]

4 Sep 2019

6. Review Comments to the Author

Reviewer #1: 1) The cut points described in lines 196 - 198 are for single observations (R = 2, MR = 4, MS = 6 and S = 8). They don't take in to account average values. What happens if the average value is 5? Will it be MR or MS? That is why you need to describe your cut off point. On the same lines (355 - 358) you described Gimiza9 with average value of 4.6 as MS. I don't believe it's MS, rather it is closer to MR than MS. Describing the cut off value will avoid such confusions.

Thank you for your comment, we added detailed description for the cutoff in the materials and methods as you suggested. 

2) Lines 201 - 202: The authors described how they calculated the ACI; but at no point did they use the ACI values in the results or discussions part. Hence, you may delete the description in the materials and methods part.

ACI description was removed from the materials and methods 

3) The introduction part is still too long. It can be significantly shortened.

Unnecessary parts were removed from the introduction 

Reviewer #2: Dear Dr Ibrahim Elbasyoni,

all comments have been addressed. Critical comments have been considered. Please change only line 107: "accession" to accessions.

Thank you so much for your comment the required correction was conducted 

 The manuscript provides now valuable information's about genotypes, molecular markers, usable for resistance breeding against yellow rust. The northern regions of Africa are effected by many abiotic and biotic stresses. Yellow rust epidemics have been observed in European countries as well, so that genotypes carrying effective resistances in this study are usable in the future as the basis for an increased resistance level of elite material.

Thank you so much for the positive comment.

---

## [Editor Report · Decision Letter 2]

9 Sep 2019

Evaluation of a Global Spring Wheat Panel for Stripe Rust: Resistance Loci Validation and Novel Resources Identification

PONE-D-19-14766R2

Dear Dr. Elbasyoni,

We are pleased to inform you that your manuscript has been judged scientifically suitable for publication and will be formally accepted for publication once it complies with all outstanding technical requirements.

With kind regards,

Dragan Perovic, Ph.D

Academic Editor

PLOS ONE

Additional Editor Comments (optional):

Dear Dr Elbasyoni,

it is my pleasure to accept your manuscript for publishing at PlosONE.

Regards

Dragan
---

## [Editor Report · Acceptance letter]

23 Oct 2019

PONE-D-19-14766R2 

Evaluation of a Global Spring Wheat Panel for Stripe Rust: Resistance Loci Validation and Novel Resources Identification 

Dear Dr. Elbasyoni:

I am pleased to inform you that your manuscript has been deemed suitable for publication in PLOS ONE. Congratulations! Your manuscript is now with our production department. 

With kind regards,

on behalf of

Dr. Dragan Perovic 

Academic Editor

PLOS ONE